# Clinical Applications of Immunotherapy for Recurrent Glioblastoma in Adults

**DOI:** 10.3390/cancers15153901

**Published:** 2023-07-31

**Authors:** Meagan Mandabach Olivet, Michael C. Brown, Zachary J. Reitman, David M. Ashley, Gerald A. Grant, Yuanfan Yang, James M. Markert

**Affiliations:** 1Heersink School of Medicine, The University of Alabama at Birmingham, Birmingham, AL 35233, USA; mkmandab@uab.edu; 2Department of Neurosurgery, Duke University, Durham, NC 27710, USA; mcb52@duke.edu (M.C.B.); david.ashley@duke.edu (D.M.A.); gerald.grant@duke.edu (G.A.G.); 3Department of Radiation Oncology, Duke University, Durham, NC 27710, USA; zjr@duke.edu; 4Department of Neurosurgery, University of Alabama at Birmingham, Birmingham, AL 35233, USA; jmarkert@uabmc.edu

**Keywords:** recurrent glioblastoma, rGBM, immunotherapy, checkpoint inhibitor, vaccine, oncolytic virus, CAR T cell, immunotoxins

## Abstract

**Simple Summary:**

There are few established treatment options for recurrent glioblastoma (rGBM). Immunotherapy, which potentiates the immune system to counter tumor growth, offers new hope for treating GBM that has relapsed after conventional therapies. The aim of this literature review is to summarize recent clinical studies of immunotherapy for rGBM, including that of immune checkpoint blockade, oncolytic virotherapy, chimeric antigen receptor (CAR) T-cell therapy, cancer vaccine and antibody-conjugated toxin. The literature search was concluded in February 2023. This review of immunotherapies provides a comprehensive overview of treatment advances, limitations in each strategy, ongoing opportunities, and preliminary correlates to survival, in order to support clinical decision-making and guide future research endeavors.

**Abstract:**

Glioblastoma (GBM) is the most common malignant primary brain tumor in adults. Despite standard therapies, including resection and chemoradiation, recurrence is virtually inevitable. Current treatment for recurrent glioblastoma (rGBM) is rapidly evolving, and emerging therapies aimed at targeting primary GBM are often first tested in rGBM to demonstrate safety and feasibility, which, in recent years, has primarily been in the form of immunotherapy. The purpose of this review is to highlight progress in clinical trials of immunotherapy for rGBM, including immune checkpoint blockade, oncolytic virotherapy, chimeric antigen receptor (CAR) T-cell therapy, cancer vaccine and immunotoxins. Three independent reviewers covered literature, published between the years 2000 and 2022, in various online databases. In general, the efficacy of immunotherapy in rGBM remains uncertain, and is limited to subsets/small cohorts of patients, despite demonstrating feasibility in early-stage clinical trials. However, considerable progress has been made in understanding the mechanisms that may preclude rGBM patients from responding to immunotherapy, as well as in developing new approaches/combination strategies that may inspire optimism for the utility of immunotherapy in this devastating disease. Continued trials are necessary to further assess the best therapeutic avenues and ascertain which treatments might benefit each patient individually.

## 1. Introduction

Glioblastoma (GBM) is the most common malignant primary brain tumor in adults, with more than 13,000 new cases diagnosed in the United States annually [1]. Despite standard therapy, including gross total resection and chemo-radiation, primary or newly diagnosed GBM (nGBM) in adults almost always recurs [2,3], forcing patients to confront recurrent GBMs (rGBMs) soon after finishing aggressive treatments for the primary tumor [4,5]. Compared to the standard modalities to treat primary GBM, established by Stupp et al. [6], there are no standard-of-care therapies for rGBM, and the median overall survival (OS), even with aggressive treatment, falls precipitously under nine months once GBM recurs [7,8]. 

Therapy resistance is a greater concern in rGBM, compared to the primary disease, as the initial treatments typically select for more heterogenous, aggressive, treatment-refractory recurrent tumors [9] that harbor chemotherapy resistance mutagenesis [10]. Even surgery itself, the cornerstone for initial treatment, has been shown to trigger therapy resistance and tumor recurrence in a preclinical model [11,12]; therefore, the role of conventional therapy for rGBM is limited in patients who have already failed initial treatment. To this end, novel immunotherapies for GBM have been in active clinical trials and offer new hope for treating adult rGBM by potentiating host anti-cancer immunity. A major challenge for glioblastoma immunotherapy is the immunologically suppressive microenvironment, the mechanisms of which, in primary GBM, have been extensively studied and reviewed [13], whereas rGBM is much less investigated, due to the more limited role of surgery and lack of tissue [14,15]. Some studies show primary and rGBM harbor similar suppressive changes, with no significant differences in immunologic features [16,17], except that rGBM contains a sustained accumulation of blood-derived macrophages [18], an increased proportion of CD8+ TILs and activated memory T cells [19].

Immunotherapies for GBM employ novel strategies to address these treatment challenges, with the goal of potentiating host anti-cancer immunity. These areas include immune checkpoint blockade, oncolytic virotherapy, chimeric antigen receptor (CAR) T-cell therapy, cancer vaccine and immunotoxins [20] (Figure 1). Phase I clinical trials involving immunotherapy are usually conducted in rGBM patients, as they do not, typically, receive concurrent chemotherapy or radiation therapy, as do nGBM patients, making it easier to identify adverse effects associated with immunotherapy, per se. The safety data in rGBM is critical for this fragile patient population who have suffered, and failed to benefit, from previous aggressive therapies. However, rGBM may be more resistant to immunotherapy than primary GBM, due to evolutionary pressures from prior treatment, the immunosuppressive effects of prior chemo-radiation, longer-term exposure to corticosteroids, declining functional level, and a worse prognosis overall. Strategies to combine different lines of synergistic immunotherapy, or to combine them with current standard therapies, have also been widely explored in rGBM [21].

Here, we provide an update on clinical trials testing immunotherapy for patients with rGBM, with a special focus on the molecular markers correlated with therapy outcome, in order to identify patients that may benefit from immunotherapy and/or to elucidate features that are associated with resistance to immunotherapy.

## 2. Materials and Methods

The literature search was initiated in December 2022, and concluded in February 2023. Included studies were clinical trials, while observational studies or case reports were excluded. The selection of studies to review were ascertained by three independent reviewers (M.M.O., Y.Y. and M.C.B.) through searching online databases, including PubMed, Web of Science, Scopes, Ovid Embase, and Ovid Medline, and were limited to adult patients with recurrent glioblastomas. Relevant articles were published from the year 2000 to February 2023. Key words for the literature search were “recurrent glioblastoma”, “progressive glioblastoma”, “relapsed glioblastoma”, plus “immunotherapy” or the specific names of immunotherapy categories, e.g., checkpoint, vaccine, virus/viral therapy. CAR T, immunotoxin. The authors manually reviewed the references listed by the articles found in the first round for additional relevant trials. Components of the studies that were investigated included survival and progression outcomes, safety, and methodology, including drug delivery and surveillance techniques. Clinical trials involving both primary and recurrent GBMs, or both adult and pediatric GBMs, were excluded, unless there was stratified data for adult rGBM patients.

## 3. Results 

### 3.1. Immune Checkpoint Blockade

Relative to other tumor types, that are more amenable to immunotherapy, the microenvironment for GBM is particularly immunosuppressive, harboring high densities of CD8 T cell-suppressive cytokines, myeloid-derived suppressive cells and regulatory T cells [22,23,24]. Cytotoxic T lymphocytes (CTLs) in GBM express multiple terminal exhaustion markers (e.g., TIM3, LAG3, PD1, CTLA-4 and TIGIT), many of which contribute to T cell dysfunction upon binding to their respective ligands (e.g., phosphatidyl serine, PD-L1, CD155), which are also present within the TME [19,25]. Immune checkpoint inhibitors (ICIs) utilize monoclonal antibodies that block the suppressive receptor:ligand interactions in T cells to prevent tumors from down-regulating CD8 T cell immunity [26,27]. Clinically explored targets of immune checkpoint blockade therapy include programmed cell death protein-1 (PD-1), programmed death ligand-1 (PD-L1) and cytotoxic T-lymphocyte-associated protein 4 (CTLA-4).

#### 3.1.1. PD-1 Inhibition

Anti-PD-1 inhibitors have been extensively studied and have shown promise in treating solid tumors, including non-small cell lung cancer (NSCLC), renal cell carcinoma (RCC), urothelial cancer and hepatocellular carcinoma [28,29,30,31]. Pembrolizumab, nivolumab (Nivo), and cemiplimab are PD-1 inhibitors approved for clinical use. PD1 blockade has been tested in rGBM with a manageable safety profile, but there has been no clear evidence of efficacy as a monotherapy. In the largest phase III randomized clinical trial conducted with ICI in rGBM to date, CheckMate 143 (NCT02017717) [32], Reardon et al. sought to compare the efficacy of bevacizumab to Nivo in 369 GBM patients at first recurrence. The primary endpoint was overall survival (OS), which was defined as the time from randomization to death from any cause. No statistical difference in the 12-month OS or median OS was observed between bevacizumab and Nivo. Sub-stratification of patients revealed that no baseline characteristics of steroid use and *MGMT* methylated tumors (a prognostic feature in GBM) were associated with better survival in the Nivo-treated arm, but not the bevacizumab arm of the trial, possibly indicating the activity of a PD1 blockade in a subset of patients.

KEYNOTE-028 was a phase Ib, multi-cohort trial to study the response rate (primary endpoint) and several secondary endpoints, including safety, OS and progression-free survival (PFS) of pembrolizumab in 26 adults with PD-L1-positive rGBM, and it showed manageable toxicity and an 8% overall response rate, although clinical response did not correlate with PD-L1 expression [33]. A multicenter, open-label, two-cohort randomized phase II study by Nayak et. al. (NCT02337491) sought to evaluate if anti-VEGF therapy could enhance Pembrolizumab efficacy in 80 patients with rGBM [34]. The primary endpoint was defined as six-month progression-free survival (PFS-6). This study showed that combination therapy improved temporary control of the disease with a PFS-6 of 26% compared to 6.7% with Pembrolizumab monotherapy; however, no enhancement of PFS or OS was observed. A single-arm open-labeled phase I study was conducted by Sahebjam et. al., (NCT02313272) with six patients, to investigate primary endpoints of safety and tolerability of bevacizumab plus pembrolizumab with concurrent radiation [35]. The safety endpoint of the study was met, and the results were encouraging in that there was some enhancement of bevacizumab efficacy by pembrolizumab.

Recent work in other tumor types determined that neo-adjuvant ICI may be more efficacious than ICI alone [36,37]. A randomized, multi-institutional clinical trial conducted by the Ivy Consortium involved 35 patients with surgically resectable rGBM, to determine the effect of neoadjuvant therapy with pembrolizumab [38]. The primary endpoints were immune response and survival. Not only was neoadjuvant pembrolizumab well tolerated, but it also led to notable improvement in overall survival (OS) (417 days vs. 228.5 days, *p* = 0.04), and progression-free survival (PFS) (99.5 and 72.5 days, respectively, *p* = 0.03). Furthermore, the treatment demonstrated enhancement of the local and systemic anti-tumor immune responses.

#### 3.1.2. PD-L1 Inhibition

PD-L1 is another immune exhaustion marker upregulated in GBM [39]. Clinically explored PD-L1 immune checkpoint inhibitors include atezolizumab, durvalumab, and avelumab. The PCD4989g (NCT01375842) study is a multicenter, phase I study testing the primary endpoints of safety and tolerability of atezolizumab monotherapy in multiple solid and hematological malignancies. In a subset of 16 rGBM patients, analyzed by Lukas et. al. [40], safety was established and, though non-significant, there was a trend of improved OS in patients with elevated peripheral CD4+ T cells and hypermutated tumor status not taking steroids at the time of atezolizumab monotherapy. A multicenter, open-label phase II study by Nayak et al. (NCT02336165) evaluated the safety and efficacy of durvalumab (MEDI4736) alone or in combination with varying doses of bevacizumab in rGBM patients that were bevacizumab-naïve or refractory [41]. Primary endpoints of this study included OS-12, PFS-6, and OS-6. Although durvalumab was well tolerated, no improvement in OS was noted with durvalumab alone or in combination; however, it was noted that an early increase of systemic Ki67+CD8+ T cells was associated with improved OS-12 and PFS-6. A stratified, open-label, phase II clinical trial (GliAvAx), by Awada et al., assessed the safety profile and secondary endpoint of efficacy for the administration of axitinib (anti-VEGF) with avelumab (anti-PD-L1) in 54 patients [42]. The results demonstrated that the safety profile was adequate; however, there was found to be no benefit of the therapy in the treatment population, or any of the subpopulations.

#### 3.1.3. CTLA-4 Inhibition

Therapeutic anti-CTLA4 antibody, ipilimumab (Ipi), potentiates cancer immunity by blocking the B7-CTLA-4 negative regulation of T cell priming during antigen presentation. Omuro et al. analyzed the exploratory phase I cohort of the CheckMate 143 study (NCT02017717), in which 40 rGBM patients were partially randomized to assess the safety profile (primary endpoint) of Nivo alone, or Nivo+ IPI combination in 1:3 or 3:1 dosage [43]. In total, 10% of the patients on Nivo alone had adverse events (AE) leading to study discontinuation, and the percentage was higher with combination therapy, while Nivo had a better safety profile compared to IPI. Thus, Nivo progressed as a therapy for the phase III portion of the Checkmate 143 trial. A phase I study by Duerinck et al. (NCT03233152), involving 27 rGBM patients, evaluated the safety and efficacy of intracerebral injection of IPI alone, or IPI plus Nivo into the tumor resection cavity, following neoadjuvant therapy with intravenous NIVO [44]. Intracerebral administration of IPI ± Nivo was deemed safe, with mild and manageable immune-related adverse events. Outcome comparison with historical control was mixed with favorable OS, but no significant difference in PFS. This suggests that intratumoral administration may be an efficacious way to improve survival, while reducing systemic toxicity.

#### 3.1.4. Ongoing Studies

Clinical trials combining Nivo and a conditionally activated IL-12 expression vector are underway. Twenty-one rGBM patients received intraoperative peritumoral injections of adenoviral vector (Ad-RTS-hIL-12 or Ad), which delivers the IL-12 transgene upon oral intake of the RTS promotor activator VDX [45]. Compared to monotherapy with Ad-RTS-hIL-12, the combination with Nivo did not confer additional toxicities [46], which supported a subsequent phase II study (NCT04006119).

A phase II study (NCT04145115) that aims to ascertain if a combination of Ipi and Nivo increases the tumor response rate, as determined by modified Response Assessment in Neuro-Oncology (RANO) Criteria, in patients with hypermutated histologically confirmed rGBM is currently ongoing (some cases are categorized under astrocytoma, IDH-mutant, WHO grade 4, due to the new WHO 2021 classification). An active multicenter, randomized, phase Ib trial (NCT04826393) is underway to determine the safety and efficacy of the drug combination of Anti-TIGIT antibody ASP8374 with PD1 inhibitor cemiplimab. The trial involved cohort 1 (safety study for ASP8374 alone) in grade IV and some grade III recurrent high-grade glioma patients, while cohort 2 (efficacy study for ASP8374 combined with cemiplimab) enrolled only rGBM patients. A phase I trial (NCT04656535) examining safety of anti-TIGIT antibody AB154 with PD-1 blockade is recruiting, and patients will be divided into surgical and non-surgical cohorts.

### 3.2. Virotherapy

The emergence of viral therapies for GBM is based on the natural neurotropism of certain viruses, including herpes simplex-1, poliovirus, parvovirus, and adenovirus. Over the years, a number of viral immunotherapies have been developed and tested clinically for primary and recurrent GBM. They employ a variety of anti-cancer strategies. Viruses that are replication-competent act as vectors to introduce specific genes into the tumor environment, such as suicidal genes, tumor suppressor genes, or immunostimulatory genes [47,48]. Viruses that are selectively replication-competent, however, act by infecting and propagating among tumor cells, inducing oncolysis. To potentiate anti-cancer immunity, some oncolytic viruses have been genetically engineered to enhance antigen presentation, or to elicit a more effective immunity against cancer cells.

#### 3.2.1. Herpes Simplex Virus-1

Herpes simplex virus-1 (HSV-1), of the Herpesviridae family, is a double-stranded DNA virus that has evolved to coexist within the human nervous system, a property exploited for targeting brain tumors. Pre-clinical studies demonstrated the potential of using a HSV mutant for GBM treatment in the 1990s [49,50]. Further genetic modifications for clinical application centered on attenuation of neuro-virulence (not causing encephalitis, etc.) related to the γ_1_34.5 gene [51], alongside the enhancement of the capacity to replicate and lyse in dividing GBM cells [52]. An HSV-1 strain, talimogene laherparepvec (TVEC, Oncovex GM−CSF, or IMLYGIC) has been FDA-approved in treating advanced melanoma [53] and is in active clinical trials for rGBM. Several generations of oncolytic HSV-1 (oHSV) strains have been tested in clinical trials, including G207, HSV1716, G47Δ, and M032.

HSV G207

HSV G207 belongs to the first-generation oHSV, engineered with a 1000 bp deletion on the diploid γ_1_34.5 gene, that disables viral replication in non-dividing cells, and, with an E. coli lacZ insertion, inactivates the UL39 gene (encoding ribonucleotide reductase subunit ICP6) as a second safety mechanism [54].

The safety profile of G207 was first demonstrated in a dose-escalating phase I trial of G207 involving 21 patients with rGBM (NCT00036699) [55]. No adverse events or toxicity ascribed to G207 were reported at a final dose of 3 × 10^9^ pfu intra-tumoral infusion.

To assess the feasibility of multiple-dose delivery, a subsequent phase Ib, open-label study (NCT00028158) was conducted at the University of Alabama in Birmingham, with 6 rGBM patients who received both pre- and post-resection G207 infusion (total dose 1.15 × 10^9^ -pfu) via a catheter that was stereotactically placed in the tumor. The study demonstrated a satisfactory safety profile with no dose-limiting toxicities, and there was promising evidence of antitumor activity associated with repeated G207 infusion into the resection cavity [56].

A third phase I, single-site, open-label trial (NCT00157703) examined the safety of G207 intratumoral infusion followed by focal radiation therapy in nine rGBM patients. The safety endpoint was met with no patients developing HSV-related adverse effects. Two patients received additional cycles of HSV therapy without severe adverse effects. Signs of efficacy were very limited, with mPFS of 2.5 months, and a median survival of 7.5 months [57].

HSV 1716

HSV1716 is another first-generation oHSV with deletions at both γ_1_34.5 loci. It is less attenuated than G207 but retains the capacity to replicate in dividing tumors while sparing the post-mitotic brain [58]. Three phase I trials were conducted in the United Kingdom with a reassuring safety profile. Rampling et al. completed the first Phase I study of HSV1716, treating nine rGBM patients with intratumoral injection of doses up to 10^5^ PFU. No adverse symptoms attributed to HSV were reported, and four patients survived beyond 14 months [59]. A second proof-of-principle study by Papanastassiou et al. involved 11 rGBM patients treated with intratumoral injection of HSV1716, followed by surgical resection. Again, no adverse effects could be attributed to HVS1716 infusion. Additionally, viral replication was detected in the tumor bed [60]. In another phase I study, six rGBM patients received surgical resection followed by HSV1716 injections to eight different sites of the resection cavity wall, demonstrating the safety and feasibility of delivering HSV1716 to the post-resection tumor bed [61], with one patient surviving up to 22 months following viral infusion, and post-treatment MRI showing continued tumor regression.

HSV G47Δ

G47Δ is a second-generation triple mutated oHSV-1 with deletions in γ34. 5, UL39 and α47 genes, designed for enhanced viral replication and immune activation through upregulated MHC class I expression [62,63]. In a recent phase I/II, single-arm study (UMIN000002661), an escalating dose of G47Δ up to 1 × 10^9^ PFU was injected twice within 2 weeks in patients with rGBM [62]. Repeated intratumoral injection of G47Δ was found to be safe with only minor adverse effects, such as headache and fever, while achieving a mOS of 7.3 months. In a subsequent phase II, single-arm trial by Todo et. al., (UMIN000015995), 19 patients with rGBM or residual GBM received up to six doses of serial G47∆ intratumoral injections [64]. The study reached an unprecedented 1 year-survival of 84.2%, with mOS being 20.2 months following serial initiation. In three patients, survival extended beyond 46 months. G47Δ has received conditional and time-limited approval in Japan. The improved outcome in the phase II study suggested multiple dosing may be the key to better virotherapy outcomes.

#### 3.2.2. Poliovirus

An attenuated poliovirus:rhinovirus chimera (PVSRIPO, for Poliovirus Sabin Rhinovirus IRES Polio ORF), was derived from the type I Sabin vaccine strain modified with the human rhinovirus type 2 Internal Ribosomal Entry Site (IRES) [65]. PVSRIPO, trade name Lerapolturev, was initially developed for its selective cytolytic activity in glioma cell lines and the fact it lacks neurotoxicity [66], due, in part, to the inability of the HRV2 IRES to recruit ribosomes in neurons [67]. However, recently it was shown that, in human glioma ex vivo slice cultures and murine models, PVSRIPO primarily infects and induces innate antiviral signaling in non-malignant cells in the tumor microenvironment to mediate antitumor efficacy [68,69]. PVSRIPO was first tested in a dose-escalation phase I study (n = 61, NCT01491893) in rGBM using a single dose delivered via convection-enhanced delivery (CED), wherein a manageable safety profile was observed and a phase II dose of 5 × 10^7^ TCID50 was identified [70]. In addition, a 20% survival rate at 36 months after PVSRIPO treatment was observed, relative to 4% in a criteria-matched historical control cohort [70]. A larger phase II multi-institutional clinical trial with CED delivery of PVSRIPO was recently completed (NCT02986178). Early results from the phase II trial suggest similar survival dynamics as those observed in the phase I trial with mOS rates at 12 months of 50% and 54%, and survival rates at 24 months of 17% and 18%, respectively [71].

#### 3.2.3. Parvovirus

The first phase I/IIa study (NCT01301430), involving parvovirus in the treatment of rGBM, utilized the oncolytic parvovirus H-1 (H-1PV). The ParvOryx01 trial was conducted as an open, non-controlled dose-escalation, single-center study with 18 rGBM patients. Patients were assigned to either intratumoral or intravenous administration of the virus. The safety profiles of both routes of administration were satisfactory with no dose-dependent side effects or dose-limiting toxicity. Secondarily, this study demonstrated that, independent of the route of administration or dose, H-1PV could cross the blood–brain barrier and confer an mPFS of 111 days and an mOS of 464 days [72].

#### 3.2.4. Adenovirus

Oncolytic adenovirus is engineered for conditional replication in glioma cells.

ONYX-015 is the first generation of attenuated adenovirus with a deletion in E1B to selectively replicate in tumor cells but is not permissive to normal cells with intact p53 pathways [73]. In a phase I dose-escalation, multi-institutional trial, 24 patients with rGBM received ONYX-015 (up to 1 × 10^10^ PFU) injected into 10 sites at the resection cavity wall. No dose-limiting toxicity was observed and the mOS was 6.2 months in this study [74]. Clinical trials of ONYX-015 were discontinued in the US after failing a phase III trial, in combination with chemotherapy, for head and neck cancers (NCT00006106).

The next-generation oncolytic adenovirus, Delta24-RGD (DNX-2401), has a 24 bp deletion in the adenovirus E1A gene and an RGD motif in fiber protein, which is selectively optimized for replication in GBM, due to commonly mutated Rb pathways, while sparing prevents post-mitotic cells [75,76,77]. The first clinical trial was a phase I, dose-escalation study in recurrent glioma, 33 being rGBM, to assess DNX-2401 intratumoral injection alone, or injection–resection–second injection into the resection cavity wall [78]. Except for mild adverse effects related to Delta24-RGD, there was no dose-limiting toxicity observed in each group, and the study achieved a durable response with 20% having a 3-year survival rate with a single injection. A second phase I study by Van Putten et al. utilized CED for continued micro-infusion and wider distribution of Delta24-RGD injected intratumorally (up to 1 × 10^11^ PFU) into 19 rGBM patients [79]. The study showed a safety profile with manageable virus-related adverse effects, including increased ICP, edema, and self-resolving. Two patients had long-term tumor regression and one was still alive eight years after virotherapy. In terms of combination therapy, a subsequent study further tested DNX-2401 in combination with IFN-γ in a Phase Ib study for rGBM, which showed no additional efficacy from IFN-γ [80].

#### 3.2.5. Newcastle Virus

Newcastle Virus (NDV), a single-stranded RNA virus, has garnered curiosity in viral oncolytic therapy. Lentogenic strains of NDV, which are not pathogenic to poultry, have also shown therapeutic potential by generating apoptosis in tumor cells [81]. They have been combined with other tumor cells as a vaccine for various cancers including GBM [82]. Freeman et al. investigated the use of an oncolytic HUJ strain of Newcastle virus, NDV-HUJ, in a phase I/II trial [83]. In the trial, 14 patients were included, with three patients achieving long term survival and there being one additional remission during maintenance dosing followed by recurrence after 3 months.

#### 3.2.6. Reovirus

The respiratory enteric orphan virus (reovirus) has evolved to colonize the human lung and gut without causing symptoms. Its serotype 3, reolysin, is particularly adapted to infect and lyse tumors that highly express ras signaling, including GBM, without further engineering. The first clinical trial was a dose-escalation study for twelve patients with histologically confirmed rGBM, who were treated with stereotactic injection of reovirus [84]. No dose-limiting toxicity was observed; however, the mOS was only 21 weeks. A second attempt utilized CED-based viral delivery over a period of 72 h, wherein grade III adverse event (seizure) was attributed to viral therapy. Again, mOS was 20 weeks and 1 in 15 patients survived beyond 2 years [85]. A further study explored using intravenously administered reovirus as neoadjuvant therapy, combined with subsequent surgical debulking and checkpoint blockade. This was a window of opportunity study to primarily confirm viral infection in resected tumors, which was indeed positive in 9/9 patients. Lymphopenia was common in all patients and severe in 6/9 patients [86].

#### 3.2.7. Retrovirus

Retroviruses are positive-sense, single-stranded RNA viruses in which the viral RNA is reverse-transcribed and incorporated into the host genome. The most well-studied retrovirus therapy for glioma is vocimagene amiretrorepvec (Toca 511). Toca 511 is a murine leukemia viral vector that inserts the yeast cytosine deaminase (CD) gene into infected malignant cells [87]. Tumors with CD genes then convert 5-fluorocytosine (FC) to 5-fluorouracil (5-FU) locally for targeted therapy [88]. Three open-label, ascending dose phase I trials in 126 patients were conducted to study Toca 511 in rGBM to demonstrate the safety profile of Toca 511 through intratumoral administration (NCT01156584), resection followed by intratumoral injection (NCT01470794), and IV administration followed by resection (NCT01985256) [89]. A randomized, open-label phase II/III trial (TOCA 5) (NCT02414165) assigned 403 patients to undergo resection, followed by intra-cavitary injection of Toca 511, followed by oral Toca Fc, or assigned them to a control. Overall, there was no significant difference in survival [88].

#### 3.2.8. Ongoing Studies

The intramural infusion rQNestin is engineered for enhanced replication in glioma, with a γ_1_34.5 copy reinserted into G207 under a synthetic nestin promotor, selectively expressed in actively mitotic cells [90]. A phase I trial is currently recruiting rGBM patients for single or multiple (up to six) doses of rQNestin intratumoral infusion guided by intraoperative MRI (NCT03152318).

C134 is an HSV-1 chimera with human cytomegalovirus (HCMV) for optimal viral translation and replication, which retains a conservative design for attenuation [91,92]. C134 has completed pre-clinical studies [93]. Before moving on to a phase I trial for rGBM (NCT03657576) and is actively enrolling patients.

The next generations of oHSV were geared towards enhanced immune stimulatory effects, while preserving attenuation from earlier generations. To this end, human interleukin 12 (IL12) was incorporated into G207 to potentiate an antitumor inflammatory response [94]. IL-12, amplified during viral replication, activates NK cells and cytotoxic T cells via IFN-γ, potentiating both innate and adaptive immune responses. M032 is currently in a phase I dose-escalating trial for rGBM and other high-grade gliomas and has completed enrolment [95]. Another phase I/II trial of M032, in combination with Pembrolizumab, is underway in patients with nGBM and rGBM. (NCT05084430).

A phase II trial for Delta24-RGD, in combination with pembrolizumab, is underway in rGBM patients with promising preliminary efficacy (CAPTIVE study, NCT02798406) [96]. There is also an additional ongoing phase I study testing DNX-2401 and TMZ combination for rGBM in Spain (NCT01956734).

### 3.3. CAR-Modified T Cell (CART)

Chimeric antigen receptor (CAR) T cells are genetically modified to recognize and attack specific tumor cells. Having shown success in the treatment of hematologic malignancy [97,98], interest has been garnered in terms of treating solid tumors [99,100], with targets including epidermal growth factor receptor variant III (EGFRvIII), interleukin-13 receptor α2 (IL13Rα2), and Erythropoietin-producing human hepatocellular carcinoma (Eph) receptor.

#### 3.3.1. CART-EGFRvIII

EGFRvIII is commonly expressed in GBM [101] and is correlated with poor prognosis [102]. O’Rourke et al. conducted the first-in-human phase I trial (NCT02209376) studying the safety of CAR-modified T cell (CART)–EGFRvIII in ten patients with EGFRvIII-expressing rGBM [103]. No dose-limiting toxicities were observed. The peak expansion of CART-EGFRvIII cells was between days three and ten, followed by a rapid decline after day 14 to undetectable levels in 30 days on flow cytometry. At the reporting of results, two subjects remained alive but had disease progression, as evidenced by imaging. Goff et al. conducted a phase I trial (NCT01454596) with 18 patients (eight recurrent, ten primary) to determine the safety profile of autologous peripheral blood lymphocytes retrovirally transduced to become CART-EGFRvIII [104]. This study demonstrated no dose-limiting toxicities until the maximum dose was reached, which constituted greater than 10^10^ cells; however, no objective therapy response was noted on MRI with a median PFS of only 1.3 months.

#### 3.3.2. CART-IL13Rα2

IL13Rα2 is a monomeric high-affinity receptor for IL-13 and is associated with worse prognosis. IL13Rα2 is differentially highly expressed in over 50% of GBMs, but not in normal brain tissue [105]. In the first-in-human trial, Brown et al. evaluated the safety and feasibility of autologous CART-IL13Rα2 in three patients (NCT00730613) [105]. Safety was established. Evidence of transient anti-glioma activity was observed with necrosis on MRI, and down-regulation of IL13Rα2+ expression in GBM, following treatment. The mean survival of 11 months after relapse was unremarkable. Further engineering rendered CART-IL13Rα2 cells resistant to glucocorticoid treatment, and it has been tested in another phase I trial involving six unresectable rGBM patients on steroid therapy. The therapy was well tolerated when given intratumorally, in combination with human IL-12, with transient ant-tumor response observed in 4/6 patients [106].

#### 3.3.3. EphA2-Redirected CAR T-Cells

Erythropoietin-producing human hepatocellular carcinoma (Eph) receptor is the largest family of tyrosine–kinase receptors and activates the downstream signaling that is involved in cellular mobility and angiogenesis [107]. While many tumor cells, including glioblastoma, express a type of Eph receptor known as EphA2, most human tissues, including the brain, do not; therefore, EphA2 is an attractive site for immunotherapy [108]. Lin et al. conducted a pilot trial (NCT03423992) with three patients receiving EphA2-Redirected CAR T-Cells at 1 × 10^6^ cell/kg [109]. This is the first-in-human trial of CAR T-cells directed againstEphA2-positive rGBM. The primary endpoint of safety was met, but no evidence of efficacy was observed, with OS ranging from 86–181 days.

#### 3.3.4. Autologous CMV-Specific T-Cell Therapy

Emerging evidence suggests human cytomegalovirus (CMV) infection is associated with GBM progression, and CMV antigens are found in GBM but not the surrounding brain tissue [110]. A phase I trial, by Schuessler et al. (ACTRN12609000338268), administered autologous CMV-specific T-cells in 19 rGBM patients. The therapy was determined to be safe and achieved disease stabilization with increased PSF [111]. The median OS of the eleven patients that received at least one infusion was 403 days. The median time to progression was 246 days. Four out of the 19 patients remained progression-free at the conclusion of the study. There was no correlation between the number of CMV-specific T cells transferred and the OS/PFS.

#### 3.3.5. HER2-Specific CART

Human epidermal growth factor receptor 2 (HER2) is a cell surface protein expressed in glioma cells and HER2-specific CART therapy has demonstrated anti-tumor efficacy in preclinical models [112]. Ahmed et al. conducted a phase I dose-escalation study (NCT01109095) with 17 patients with rGBM (10 adult and 8 pediatric) [113]. In terms of safety, no dose-limiting toxicity was observed. The mOS was 11.1 months following therapy administration.

#### 3.3.6. Ongoing Studies

Among ongoing studies, NCT04003649 is a phase I trial to examine IL13Rα2-CAR T cells when given as monotherapy or in combination with Nivo and Ipi for rGBM. An additional phase I (NCT04214392) study is recruiting MPP2+ rGBM patients to undergo CART therapy with a chlorotoxin tumor-targeting domain. A pilot study (NCT02937844) is underway with autologous CAR T-cells that are redirected to PD-L1. The recruitment status of this study is unknown with the last known update posted in 2016. Patients are also being recruited to a phase I trial (NCT03389230) that will determine the safety of memory-enriched T cells with HER-2 specific CAR optimized with 41BB co-stimulation.

### 3.4. Immunotoxins

Immunotoxin therapy utilizes antibody-drug conjugates to bind to specific targets, which allows for the binding and internalization of the therapy. Following the uptake of the therapy in the cells, cytotoxins are released that result in cell death.

#### 3.4.1. MDNA55

MDNA55 is an engineered immunotoxin with permutated IL-4 conjugated to a modified pseudomonas exotoxin (PE), that targets glioma cells and immune suppressive cells highly expressing IL-4R. In a Phase II single-arm, open-label, multicenter study, 47 patients received MDNA55-05 via CED (NCT02858895) with the primary objective of determining radiographic therapeutic response through RANO, iRANO and mRANO criteria [114,115]. The results demonstrated that MDNA55 is at least comparable in efficacy to other common rGBM therapies. Furthermore, inadequate response to MDNA55 could be linked to inadequate tumor penetration.

#### 3.4.2. Depatuxizumab Mofodin

Depatuxizumab Mofodin (Depatux-M) is another conjugate immunotoxin, based on ABT-806, antibody targeting EGFR, which is highly amplified in GBM cells. Lassman et al. (NCT01800695) conducted an open-label Phase I study for 60 adult patients with EGFR amplified nGBM or rGBM, who received Depatux-M [116]. In terms of primary endpoint, Depatux-M had a satisfactory adverse event profile except for frequent ocular side effects. Overall, the trial showed encouraging anti-tumor activity with an objective response rate of 14.3%. Van Den Bent et al. (NCT02343406) conducted the INTELLANCE 2/European Organization for Research and Treatment of Cancer (EORTC) 1410 phase II study, a multicenter three-arm comparative, randomized trial with 260 patients with EGFR-amplified rGBM, utilizing Depatux-M with or without TMZ [117]. This study included patients at their first recurrence following chemo-irradiation with TMZ, and the primary endpoint was survival. The results demonstrated an improved overall survival in the combination therapy group with an OS of 19.8% (95% CI: 12.2, 28.8), compared to an OS of 5.2% in the control group (95% CI: 1.7, 11.7) and 10% in the monotherapy group (95% CI: 4.8, 17.6).

#### 3.4.3. Pseudomonas Exotoxin A

Cintredekin besudotox (CB), is another recombinant immunotoxin in which the truncated Pseudomonas exotoxin A (PE38QQR) is conjugated to human IL-13. A randomized phase III trial by Kunwar and colleagues sought to determine the efficacy of cintredekin besudotox (CB) via CED. In this study, 296 rGBM patients received either IL13-PE38QQR or Gliadel wafers (GW). There was no difference in safety between the two groups, and there was no survival advantage conferred by IL13-PE38QQR [118].

#### 3.4.4. Ongoing Studies

D2C7-(scdsFv)-PE38KDEL is a recombinant pseudomonas exotoxin-A conjugated to antibody targeting both the wildtype EGFR and mutant EGFRvIII, both of which are highly expressed in glioblastomas [119]. Phase I trials involving D2C7 in combination with CD40 antibody (NCT04547777) or Atezolizumab (NCT04160494), are currently recruiting patients with rGBM.

### 3.5. Cancer Vaccines

#### 3.5.1. ERC1671

ERC1671 is a cell-based vaccine using irradiated/inactivated GBM cells and their lysates from both allogeneic and autologous tumors. It is injected subcutaneously with cyclophosphamide and granulocyte–macrophage colony-stimulating factor (GM-CSF) to enhance immune priming. A double-blinded, randomized, Phase II study, comparing this combined regimen and bevacizumab to placebo plus bevacizumab was conducted in bevacizumab-naive rGBM patients (NCT01903330) [120]. The safety profile was satisfactory with the majority of grade three reactions being headaches and no higher-grade responses. Due to a favorable median OS (12.1 months), compared to the placebo (7.6 months) found during the interim analysis, the FDA issued an early termination recommendation and that a Phase III study be conducted instead, to expedite the potential filing of a new drug application (NDA) for this therapy combination.

#### 3.5.2. Dendritic Cell Vaccines

Dendritic cells (DCs) loaded with neoantigens can efficiently prime the immune system for a robust adaptive immune response against tumor cells. An Italian Phase I-II study analyzed 20 patients with IDH1 wt rGBM, who received DC loaded with autologous tumor lysates, in combination with either TMZ or tetanus toxoid pre-conditioning. DC therapy was relatively well tolerated with seizure being the most severe adverse effect (in 3 patients). A cohort with tetanus toxoid combination showed a survival benefit compared to TMZ combination [121]. Hu et al. conducted an open-label, single-institution phase I study using autologous DC vaccine pulsed with lysate from a GBM stem-like cell line in 25 rGBM and 11 nGBM patients (NCT02010606) [122]. Looking at stratified data among rGBM patients alone, there were only grade 1 and 2 adverse events attributable to the vaccination. Although not powered to assess efficacy, the rGBM group reached a favorable PFS-6 of 24% and an mOS of 13.2 months, demonstrating evidence of survival benefits, compared to historical controls. In 2023, Liau et. al. reported positive results from a Phase III study using a DC vaccine loaded with autologous GBM lysates (DCVax-L) [123]. Patient cohorts with both nGBM and rGBM significantly improved in mOS. The rGBM patients had a 42% reduction in relative risk of death and a long-term (30 month) survival of 11.1%, compared to 5.1% in matched controls.

#### 3.5.3. Wilms Tumor 1 (WT1) Peptide Vaccination

The WT1 gene is overexpressed in a variety of solid tumors, including GBM [124]. A WT1 protein-derived peptide vaccine (DSP-7888) has completed a phase I clinical trial in a variety of advanced malignancies, including recurrent glioblastomas. (NCT02498665) A phase II study by Izumoto et al. enrolled 25 patients with WT1/HLA-A*2402-positive rGBM [125]. The protocol was well tolerated and obtained a PFS-6 of 33.3%.

#### 3.5.4. Personalized Peptide Vaccination (PPV)

Personalized peptide vaccination (PPV) takes into account the diversity of neoantigens and immunological responses among patients to optimize immunogenicity and reduce adverse reactions. Narita et al. conducted a randomized, double-blind, phase III trial comparing PPV with a placebo control in 88 rGBM patients [126]. Unfortunately, the study failed to demonstrate survival and clinical benefits for rGBM. Efficacy was comparable to the placebo group. For patients that received the PPV, the benefit in median OS was insignificant compared to the placebo.

#### 3.5.5. Heat–Shock Protein Peptide Complex-96 

Bloch et al. examined a heat–shock protein peptide complex-96 (HSPPC-96) vaccine in a phase II study (NCT00293423) among 41 patients with surgically resectable rGBM, with the primary endpoint of overall survival at six months [127]. The median OS was 42.6 weeks with a promising 6-month OS of 90.2% and a 12-month OS of 29.3%. There was only one grade 3 adverse event associated with the vaccine.

#### 3.5.6. Survivin Long Peptide Vaccine (SurVaxM)

Survivin is an inhibitor of apoptotic peptides that confers a survival advantage to tumor cells [128]. The survivin long protein vaccine (SurVaxM) contains a segment of the survivin protein and was studied in a phase I trial for nine patients (NCT01250470) with surviving positive rGBMs, who also carried HLA-A*02 or HLA-A*03 MHC class I alleles [129]. No significant adverse effects were elicited, with fatigue and localized erythema at the injection site being common. The mOS was 86.6 weeks and the mPFS was 17.6 weeks. One patient achieved complete remission 174 weeks following vaccine administration.

#### 3.5.7. EO2401

EO2401 is a microbiome-derived vaccine that is being examined in an ongoing phase I/II (NCT04116658), the results of which have been published in a preliminary abstract [130]. A total of 40 patients were treated and assigned to monotherapy or combination therapy with Nivo or bevacizumab. The vaccine was well tolerated across treatment groups. The mPFS was 1.8 months at 6 months and 14.7 months at 12 months. The full manuscript and reporting of results for this trial are in progress.

#### 3.5.8. VXM01

VXM01 is a vaccine consisting of attenuated Salmonella typhi Ty21a carrying a plasmid which encodes for vascular endothelial growth factor receptor (VEGFR)-2. A phase I study by Wick et al. (NCT02718443) utilized this vaccine in 14 patients [131]. The vaccine was shown to be well tolerated and one patient demonstrated a response to the therapy.

#### 3.5.9. Ongoing Studies

DSP-7888 (see Section 3.5.3) is currently being tested in a randomized, multicenter, adaptive Phase III Study involving 236 rGBM patients to access its therapeutic efficacy in combination with Bevacizumab (NCT03149003). A phase I study (NCT03360708) examining safety for an allogeneic tumor lysate-pulsed autologous dendritic cell vaccine is active with 20 rGBM patients. Another just-completed phase II trial (NCT01814813) involved 90 rGBM patients to assess the HSPPC-96 vaccine, given in combination with bevacizumab, completed in May 2023. A phase I trial (NCT04201873) is recruiting patients to assess the pharmacodynamic profile of the combination of pembrolizumab with an autologous tumor lysate-pulsed dendritic cell vaccine. Though the results are not publicly available, a phase I/II open-label trial (NCT01081223), that enrolled 14 rGBM patients to undergo vaccination with TVI-Brain-1, has been completed. A completed phase I trial (NCT00890032) with 50 rGBM patients was conducted to determine the safety and efficacy of a dendritic cell vaccine loaded with an autologous brain tumor stem cell mRNA. A multi-center randomized controlled phase III trial (NCT04277221) is undergoing recruitment to determine the efficacy of an autologous dendritic cell/tumor antigen (ADCTA-SSI-G1) vaccine for adjuvant immunotherapy in rGBM.

### 3.6. Future Directions: Overcoming Obstacles to Successful Cancer Immunotherapy in rGBM

Targeting brain tumors with systemic and local immunotherapies comes with unique challenges, due to the location and nature of GBM [132] (see Figure 2 for an overview).

#### 3.6.1. The Blood–Brain Barrier

While the previously held view, that the blood–brain barrier (BBB) unequivocally prevents access of the peripheral immune system (immune privilege) to the brain, has been refuted by recent works, including the discovery of CSF drainage to the cervical lymph nodes [133,134,135,136], it remains a hurdle for systemic immunotherapy delivery. The extent to which the BBB restricts T cell and immunotherapy drug access in GBM remains to be determined, particularly in rGBM after surgical resection and standard-of-care therapy. A common route to mitigate BBB restrictions on drug/T cell trafficking to the tumor, that has been employed by several aforementioned strategies (e.g., T cells, immunotoxins, and viruses), is that of intratumoral delivery. This includes delivery of CAR T cells infused intracranially [106]. Other strategies are also being developed to breach the BBB in GBM patients in order to enhance drug delivery and efficacy; for example, ultrasound disruption via an intratumor-implanted device to locally compromise the BBB in the tumor bed (e.g., NCT04528680) [137] Nonetheless, recent studies demonstrate that the BBB does not preclude the activity of PD-1 blocking antibodies at the site of the tumor, as pre- and post-treatment analyses of rGBM tumor tissue revealed changes in both DCs and T cells after neoadjuvant PD-1 blockade [138].

#### 3.6.2. Intra-Patient and Inter-Patient Heterogeneity

Gliomas exhibit notorious histological and molecular heterogeneity both within and between patients, rendering even accurate grading of GBM difficult. For this reason, efforts to define specific subtypes of brain tumors using molecular definitions [139], including recent work to use DNA methylation status for diagnosis and classification, [140], and for clinical decision making are ongoing. Moreover, transcriptional subtypes (e.g., mesenchymal, classical, and proneural) have been shown to not only vary between patients [141], but also within the tumors of individual patients [142]. Diverse neurodevelopmental transcriptional states within the same tumors are also evident [143]. The clonality and evolution of the mutational and epigenomic landscapes also vary during standard-of-care in GBM [144,145,146]. It remains to be determined how heterogeneity impacts immunotherapy susceptibility in GBM; however, the evolutionary plasticity afforded by heterogeneity is regarded as one of the most formidable barriers to immunotherapy success [147,148,149]. Potential routes to overcome heterogeneity include combining therapies to restrain tumor evolution, better-defining patient-specific features associated with therapy response (i.e., ‘precision medicine’), and elucidating mechanisms of immunotherapy resistance in clinical trial-associated analyses.

#### 3.6.3. Microglia/Myeloid Cell Immunosuppression

Upwards of 50% of the tumor mass in GBM is that of myeloid cells and microglia, widely considered dominant mediators of T cell suppression and immunotherapy resistance [150]. The majority of myeloid cells in the GBM tumor microenvironment are that of bone marrow-derived macrophages with gene expression profiles reflecting a diverse array of inflammatory states [151]. Despite the vast number of known mechanisms by which myeloid cells suppress antitumor T cell immunity, the extent to which myeloid cells explain the lacking efficacy of immunotherapy in GBM remains to be determined. Amongst the strategies being developed to target the exaggerated myeloid cell presence in brain tumors is that of myeloid cell depletion (e.g., CSF-1R blocking antibodies and inhibitors), re-education or reprogramming of immunosuppressive myeloid cells (e.g., with macrophage targeting viruses, pattern recognition receptor agonists, or antibodies) [152,153,154,155], and leveraging the robust chemokine gradient attracting bone marrow-derived macrophages in GBM for therapy with engineered Chimeric Antigen Receptor macrophages that induce immunogenic phagocytosis upon recognition of tumor antigens [156]. Strategies to target the phagocytosis inhibitory receptors/ligands (e.g., CD47 and SIR-1a) to promote microglia/myeloid cell activation are also being developed to target this compartment [157].

#### 3.6.4. T Cell Suppression

The paucity of T cells in the GBM microenvironment has long been noted, even in the absence of immunosuppressive steroid treatment. Recent work has demonstrated that the limited number of T cells infiltrating GBMs express terminal exhaustion markers [19,25], consistent with a lack of functionality. A proportion of T cells infiltrating brain tumors are immunosuppressive regulatory T cells, that actively suppress the function of antitumor T cells [158]. A level of dysfunction, potentially specific to brain tumors, is that of T cell sequestration and peripheral immune dysfunction, which may represent a mechanism of CNS-mediated immunosuppression [159,160,161]. Further studies are needed to define and overcome glioma specific routes of T cell dysfunction. It also remains to be determined whether adoptive T cell therapies (e.g., CAR T cells) will, similarly, be suppressed, or whether they can be engineered/modified to overcome such suppression. Potential strategies to subvert T cell suppression include the following: combination immunotherapy strategies, as suggested by the induction of other immune checkpoints on T cells after PD1 blockade in rGBM [138]; timing surgical resection to alleviate GBM mediated suppression [38]; and/or empowering antitumor immunity via peripheral or local antitumor vaccination.

## 4. Discussion

Novel treatments for rGBM are urgently needed. Immunotherapy has revolutionized the treatment of both hematologic and solid tumor malignancies. To date, immunotherapy has shown some promising, albeit limited, potential to prolong the survival of patients with rGBM.

ICI has shown some activity in subsets of rGBM patients, but trials to date reveal that the durable responses seen with monotherapy ICI in melanoma and other immunogenic cancer types are not likely to be achieved in rGBM. Of note, a transient improvement in OS and PFS was observed in PD-1 blockade, in combination with Pembrolizumab, at 6 months. When comparing total OS and PFS, however, there was no significant improvement. Similar findings were seen within the evaluation of durvalumab, alone, or in combination with varying doses of bevacizumab. Although no improvement in OS was found with either monotherapy or combination therapy, there was an early increase of systemic Ki67+CD8+ T cells, associated with improved OS at 12 months and PFS at 6 months. These and other recent studies suggest that ICI positively impacts T cell function [138]. Further investigation may reveal the occult mechanism that results in only a brief respite for patients.

In conjunction with a seemingly temporary response to therapy, combination therapy has shown some promise in improving survival. The phase II study by Nayak (NCT02337491) demonstrated that pembrolizumab combination therapy was more effective at 6 months compared to the monotherapy arm. Additionally, pembrolizumab was found to enhance the efficacy of bevacizumab (NCT02313272). Future studies aimed at combining various methods of treatment may reveal a favorable therapy that takes advantage of multiple modalities. In addition, ongoing efforts are seeking to determine if modifying the sequence of therapy, e.g., prior to surgery, may have added benefits.

Sequencing and combination of emerging immunotherapies with radiation therapy also merit consideration. Since most GBM patients receive standard-of-care radiation therapy in the nGBM setting, the immune state of rGBMs that have uniformly received prior radiation therapy merits careful consideration. Focal reirradiation is a common treatment option for patients with rGBM. Preclinical studies suggest that radiation therapy acts synergistically with several immunotherapeutic approaches. For instance, radiation therapy may increase HSV spread throughout the tumor and enhance replication two- to five-fold without additional morbidity or mortality [162,163,164]. A single low dose (5 Gy) delivered 24 hours after oHSV was optimal, and equally effective as that of higher doses [164]. This approach was well-tolerated and exhibited responses in a small study of nine patients conducted on patients with recurrent GBMs treated with G207 [57]. Potential mechanisms of HSV and radiation therapy synergism include radiation-induced transcription of cellular genes that complement viral gene deletions [164,165], induction of the DNA damage response in a manner that favors viral replication [166], and/or enhancement of innate and adaptive immune responses that increase oncolysis via increased tumor antigen presentation, chemokine induction, and effector T cell recruitment [167]. In the Phase I trial of G207 in pediatric rGBM, correlative analyses suggested that low-dose radiation therapy stimulated recruitment of CD4+ and CD8+ TILs [168]. Further work will be needed to identify appropriate combinations, radiation dosing and fractionation, and sequencing between radiation therapy and the most promising emerging immunotherapy agents.

The safety of intratumoral administration of various therapies has been thoroughly demonstrated. A potential concern from a viral therapy perspective, particularly when utilizing neurotropic viruses such as HSV, is the potential for viral encephalitis. Of the published trials included in the review, no patients experienced encephalitis. Encouraging safety profiles were noted across all the modalities explored in this review, suggesting that toxicity from these potential therapies should not be a barrier for continued research efforts. Furthermore, the unprecedented success of Todo and colleagues (UMIN000015995) with a one-year survival of 84.2% following the repeated intratumoral administration of G47Δ suggests that multiple dosing is safe and may contribute to better outcomes. Whether oncolytic virotherapy will mediate durable benefits in larger, randomized cohorts of GBM patients remains to be determined. Ongoing efforts are focused on optimizing viral vector design, and delivery (including frequency of treatment), as well as rational combination strategies to enhance therapeutic effect.

Lastly, the development of antigen-targeted approaches, including immunotoxins, CAR-T cells, and cancer vaccines for rGBM is a growing area of research, with continued advances in optimizing the choice of glioma selective targets, the formulation and manufacturing processes, delivery routes, and combination approaches. To date, these modalities have demonstrated clinical feasibility, and some have moved to the newly diagnosed GBM setting, e.g., including tumor lysate-loaded DC vaccines, that showed favorable results in a phase III clinical trial relative to contemporaneous matched external controls [123].

Mechanisms of action for immunotherapy modalities tested in rGBM patients covered in this review. Summarized mechanisms of action for each modality are shown in Figure 1. Immune checkpoint inhibitors (ICI) seek to revive T cell function by blocking the signaling of suppressive ‘checkpoints’, and their ligands on T cells. Virotherapy harnesses the selective oncolytic capacity of attenuated viruses in glioma cells to mediate direct killing of glioma cells, which is a consequence of the viral replication cycle, while also engaging innate inflammation in cells within the tumor microenvironment: either via direct infection, the release of immunogenic features from lysed cells, or both. Immunotoxins leverage the specificity of engineered antibodies to tumor-associated antigens to deliver highly toxic payloads (e.g., pseudomonas toxin). Chimeric Antigen Receptor (CAR) T cells are engineered autologous T cells genetically modified to express an antibody specific to a glioma-associated antigen (or antigens) that triggers (or trigger) activating signaling to enable trafficking to the tumor site, and direct killing of cancer cells via the engineered T cells. Cancer vaccines target tumor-associated antigens are delivered in the form of immunogenic peptides, protein complexes, RNA, and/or antigen pulsed dendritic cells, and seek to expand pools of tumor antigen specific antitumor immunity (Table 1).

## 5. Conclusions

Immunotherapy has been extensively studied for the treatment of rGBM. Treatment continues to be challenging, with some promising results in early trials and the later iterations of studies often having less-compelling outcomes. This review of rGBM immunotherapy for adult highlights advances and setbacks in immune checkpoint blockade, immunotherapy, CART, cancer vaccine, and immunotoxins. Globally, these trials highlight the immense work that remains to be done to bring effective care to rGBM patients. Despite the fact that a clear breakthrough has not yet occurred, there are signs of progress, particularly in the area of immune checkpoint blockade and viral therapy, combination therapy, and the use of intratumoral administration. Future studies aimed at further exploration of these areas are needed.

## Figures and Tables

**Figure 1 cancers-15-03901-f001:**
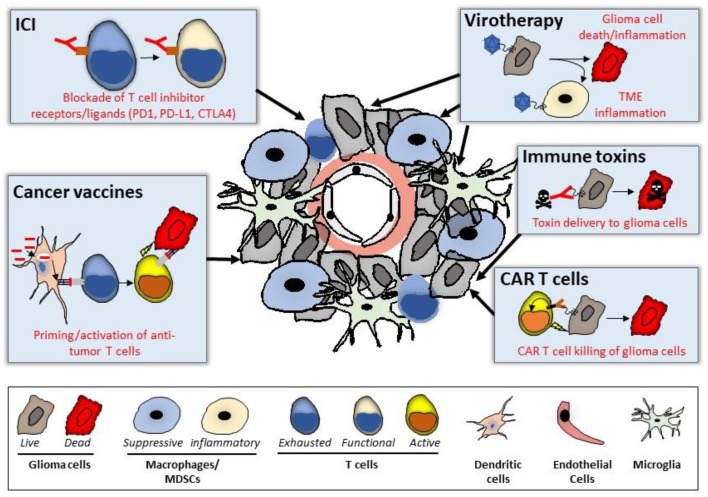
Summary of modalities tested in recurrent GBM patients with principal mechanisms of action.

**Figure 2 cancers-15-03901-f002:**
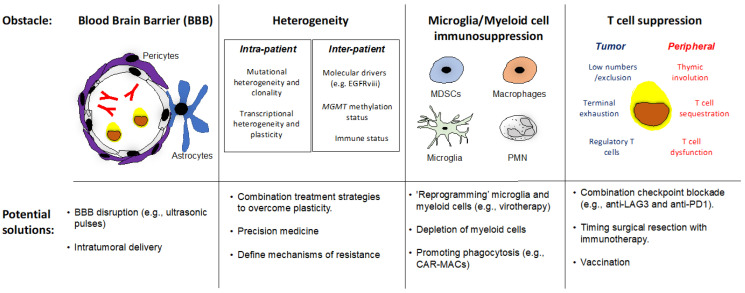
Hurdles to successful immunotherapy in rGBM. The blood–brain barrier (BBB), glioma heterogeneity, glioma-associated microglia/myeloid cells, and T cell suppression are established features in GBM that are anticipated to resist immunotherapy. See Section 3.6 for discussion. MDSCs = Myeloid Derived Suppressor Cells, PMN—polymorphonuclear leukocytes, *MGMT* = O-6-mehtyguanine-DNA methyltransferase; CAR = Chimeric Antigen Receptor.

**Table 1 cancers-15-03901-t001:** The Most Advanced Phases of Clinical Trials for Recurrent Glioblastoma in Each Category of Immunotherapy.

Therapy		Highest Phase Achieved	Trial
Immune Checkpoint Blockade	PD-1	III	CheckMate 143
	PD-L1	II	NCT02336165, GliAvAx
	CTLA-4	I	CheckMate 143
Herpes Simplex Virus	G207	IIb	NCT00028158
	1716	I	NCT02031965
	G47Δ	II	UMIN000015995
Poliovirus	PVSRIPO	II	NCT02986178
Parvovirus	H-1PV	I	ParvOryx01
Adenovirus	ONYX-015	I	Chiocca et al. [45]
	Delta24-RGD	Ib	Lang et al. [75]
Newcastle Virus	NDV-HUJ	II	Freeman et al. [83]
Reovirus	Reolysin	I	2011-005635-10
Retrovirus	Toca 511	III	NCT02414165)
CART	EGFRvIII	I	NCT02209376, NCT01454596
	IL13Rα2	III	Kunwar et al. [118]
	EphA2-Redirected	Pilot	NCT03423992
	Autologous CMV	I	ACTRN12609000338268
	HER2	I	NCT01109095
Immunotoxins	MDNA55	II	MDNA55-05
	Depatux-M	II	INTELLANCE 2/EORTC 1410
	Pseudomonas exotoxin A	III	PRECISE Trial NCT 00076986
Cancer Cell Vaccine	ERC1671	II	NCT01903330
	Dendritic cell	III	NCT00045968
	WT1	II	Izumoto et al. [25]
	PPV	III	Narita et al. [126]
	Heat-shock protein peptide complex-96	II	NCT00293423
	SurVaxM	I	NCT01250470
	EO2401	II	NCT04116658
	VXM01	I	NCT02718443

## Data Availability

Not applicable.

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
