# Peer review of "Clinical Applications of Immunotherapy for Recurrent Glioblastoma in Adults"

_cancers, 2023, doi:10.3390/cancers15153901_

Round 1
Reviewer 1 Report
In this review article, the authors summarize clinical studies of immunotherapy for recurrent glioblastoma. This is based on studies identified by a literature search of three independent reviewers during December 2022 - February 2023. The immunotherapeutic strategies that are summarized comprise immune checkpoint blockade, oncolytic virotherapy, chimeric antigen receptor (CAR) T-cell therapy, cancer vaccine and antibody-toxin conjugates.
This review is written very well and provides a good overview and useful information.
Minor
Although partially mentioned, the authors could add optionally a small section of “emerging immunotherapeutic strategies” and for instance describe here separately the cytokine-based immunotherapies, e.g., the studies by Chiocca et al. (Chiocca et al. Regulatable interleukin-12 gene therapy in patients with recurrent high-grade glioma: Results of a phase 1 trial. Sci Transl Med. 2019 Aug 14;11(505):eaaw5680. doi: 10.1126/scitranslmed.aaw5680. PMID: 31413142; PMCID: PMC7286430, or Chiocca et al. Combined immunotherapy with controlled interleukin-12 gene therapy and immune checkpoint blockade in recurrent glioblastoma: An open-label, multi-institutional phase I trial. Neuro Oncol. 2022 Jun 1;24(6):951-963. doi: 10.1093/neuonc/noab271. PMID: 34850166; PMCID: PMC9159462) or also the study of Weiss et al. (Weiss T et al. Immunocytokines are a promising immunotherapeutic approach against glioblastoma. Sci Transl Med. 2020 Oct 7;12(564):eabb2311. doi: 10.1126/scitranslmed.abb2311. PMID: 33028706). However, even without this, the study is sufficient for publication in Cancers.
Author Response
We appreciate the thoughtful comments and recommendations in your review. We have revised the manuscript based on your opinion. Please see below for a point-by-point response:
Question 1: “Although partially mentioned, the authors could add optionally a small section of “emerging immunotherapeutic strategies” and for instance describe here separately the cytokine-based immunotherapies…”
Answer 1: We appreciate the reviewer’s suggestion and agree with the importance of the work mentioned. We did highlight the phase I work with Ad-RTS-hIL-12 (Chiocca et al. Sci Transl Med. 2019; Chiocca et al. Neuro Oncol. 2022) in section 3.1.4 as recommended by the reviewer. However, given the recommendations from other reviewers to strictly focus on clinical trials with reported endpoints, we are hesitant to add additional references on preclinical work unfortunately.
Reviewer 2 Report
The authors did a large review of immunotherapy approach in recurrent Glioblastoma (rGB).
Based on what declared in the introduction, I would suggest to strictly focus on clinical trial methodologies and results, nd for all the trials the authors mentions to declare at least primary end points and results. Please further specify the search criteria that the authors used (for exemple: what they meant by “specific names of immunotherapies.”?).
I would exclude case reports and avoid considerations on the mechanisms of action/response of the therapy (e.g.3.1.5 or 3.6), which are not adherent to the focus of the work.
Overall, to make the work more understandable, I Would summarize the trials in tables.
Please,
a. be consistent to what declared in inclusion/exclusion criteria (only adult rGB): then, do not report trials studying: 1- pediatric cases (e.g. NCT04323046), all recurrent glioma and not exclusively rGB (e.g: NCT04145115, NCT04826393, NCT02303678) or trials that studying newly as well as rGB at the same time (e.g. NCT02010606), unless specific data (for nGB and rGB) are available.
In the section 3.5.2 dendritic cell vaccine include also doi: 10.1093/noajnl/vdz022.
b. Structure the paper to get a clear destinction between the concluded trials and the ongoing (I would specify that the authors titled the chapter 3.5.9 as “ongoing studies” but mentioned, e.g., NCT03149003 in another paragraph)
Minor reviews:
-use GB for glioblastoma instead of GBM
- use references more updated for the first paragraph
-please, rephrase: “Clinical trials involving immunotherapy often start with rGBM patients, as there is typically no standard-of-care concurrent chemotherapy or radiation as in primary GBM patients, making it easier to analyze the safety and clinical profiles.” (pag2, row 70-71).
Based ot the above observations, the paper need to major review.
Author Response
We appreciate the thoughtful comments/recommendations and have revised the manuscript based on the reviewer’s opinion. Please see below for a point-by-point response:
- “Please further specify the search criteria that the authors use”
Authors' reply: We agree with the reviewer’s comment and have further specified the key words and methodology used during the literature search in the method section.
- “Exclude case reports and avoid considerations on the mechanisms of action/response of the therapy (e.g.3.1.5 or 3.6), which are not adherent to the focus of the work”
Authors' reply: We agree with the reviewer that mechanisms of action/response to therapy are not the direct focus of the work. We have removed section 3.1.5 from the text, as that work has been reviewed by others elsewhere. However, we feel that given the lacking efficacy signals and overall disappointing progress applying immunotherapy to rGBM to date, combined with the goal of this work to accurately present the state of immunotherapy in rGBM demands some discussion on why immunotherapy has failed to show any clear benefit in rGBM, as well as efforts to overcome obstacles to immunotherapy success (section 3.6). Given the expertise of several of the authors in this area, the paradigm shift in the GBM community to focus on drug mechanisms of action and resistance in more mechanistically focused clinical trials to address these issues, as well as the need to reflect the substantial ongoing preclinical and early-stage clinical progress applying immunotherapy in GBM, we feel inclusion of section 3.6 is required to fully present the current state of the field. We are willing to consider further if the reviewer and/or editor strongly feels it detracts from the utility and presentation of the work.
- “Overall, to make the work more understandable, I would summarize the trials in tables.”
Authors' reply: We agree with the reviewer’s comment that case reports of occasional long-survivor in early phase trials are less important than primary and secondary outcome. So we updated the manuscript to merely focus on primary outcome without mentioning single cases. We have included Table 1 that summarizes leading clinical trials covered in this work.
- “Be consistent to what declared in inclusion/exclusion criteria (only adult rGB): then, do not report trials studying: 1- pediatric cases…, all recurrent glioma and not exclusively rGB … or trials that studying newly as well as rGB at the same time”
Authors' reply:
- We agree with the reviewer’s comment and have added specific details in the inclusion/exclusion criteria section. Studies involving both new and recurrent, or both adult and pediatric glioblastomas are excluded unless there is stratified data specifically for adult recurrent glioblastoma.
- NCT04323046 trial involved both pediatric and young adult (6m ~ 22yr) with no stratification data, thus was excluded from the review.
- NCT04145115 trial (https://classic.clinicaltrials.gov/ct2/show/NCT04145115) started in 2019, including only histologically confirmed recurrent glioblastoma (WHO IV) in adults. However, due to the new diagnostic language change under WHO 2021 classification, some patients were re-grouped under “Astrocytoma, IDH-mutant CNS WHO grade 4” under the new language. Since all other studies reviewed here were conducted before 2021 under histological classification (consistent with NCT04145115 trial), we still included the study, but added with a sentence stating the fact that the inclusion criteria is consistent with other trials reviewed here.
- NCT04826393 trial involved histologically confirmed WHO grade IV glioblastoma or its variants. Some cases with grade III recurrent malignant glioma were allowed to enroll in Cohort 1 (safety study) only, but not in Cohort 2 (efficacy study). So, we clarified in the text that Cohort 2 enrolled exclusively grade IV glioblastoma by histologic classification.
- NCT02303678 is recruiting patients with recurrent high-grade glioma (both grade III and grade IV) with no clear indication of stratification, thus was removed from the ongoing studies section. NCT02010606 involved 11 nGB and 25 rGB with stratification of data. Therefore, we clarified in the text to only discuss the data specific for rGB.
- “In the section 3.5.2 dendritic cell vaccine include also doi: 10.1093/noajnl/vdz022.”
Authors' reply: We agree with the comment and included this relevant study.
- “Structure the paper to get a clear distinction between the concluded trials and the ongoing (I would specify that the authors titled the chapter 3.5.9 as “ongoing studies” but mentioned, e.g., NCT03149003 in another paragraph)”
Authors' reply: We very much appreciate the reviewer pointing this out and we have revised the manuscript to ensure that all ongoing trials are described in respective ‘ongoing studies’ sections, including the referenced example.
- “Use GB for glioblastoma instead of GBM”
Authors' reply: We agree that GB is more intuitive than GBM, and that the field should make this change after ‘glioblastoma multiforme’ was renamed ‘glioblastoma’. However, unfortunately to our knowledge glioblastoma is still most commonly referred to as ‘GBM’ in all literatures reviewed here. For this reason, unless directed by the editor or pointed towards information suggesting contrary, we have left it as ‘rGBM’ or ‘GBM’ in the text.
- “Use references more updated for the first paragraph”
Authors' reply: Thank you for this comment. The use of some older literature in the introduction is due to our citing seminal work outlining standard of care for GBM and the background of this review.
- “Please, rephrase: “Clinical trials involving immunotherapy often start with rGBM patients, as there is typically no standard-of-care concurrent chemotherapy or radiation as in primary GBM patients, making it easier to analyze the safety and clinical profiles.” (pag2, row 70-71).”
Authors' reply: Thanks for the comment and we have rephrased this point in the original manuscript. “Phase I clinical trials involving immunotherapy are usually conducted in rGBM patients, as they don’t typically receive concurrent chemotherapy or radiation therapy as in nGBM patients, making it easier to identify adverse effects associated with immunotherapy per se.”
- “Lines 380 and 386. ‘Delta24-RGV’ should be written as ‘Delta24-RGD’”
Authors' reply: This change has been made.
- “Lines 804 and 805. The abbreviations should be placed immediately below Figure 2”
Authors' reply: We have added definitions for the abbreviations at the end of the Figure 2 legend. Thank you for the suggestion.
Thank you for your consideration, and for the thoughtful suggestions.
Reviewer 3 Report
This high quality review article provides good insight into the progress of immunotherapeutic approaches in clinical trials.
Minor concerns:
Lines 380 and 386. “Delta24-RGV” should be written as “Delta24-RGD”
Lines 804 and 805. The abbreviations should be placed immediately below Figure 2
Author Response
We appreciate the thoughtful comments and recommendations in your review. We have revised the manuscript based on your opinion. Please see below for a point-by-point response:
- “Lines 380 and 386. ‘Delta24-RGV’ should be written as ‘Delta24-RGD’”:a. This error has been corrected both in the text and table. Thanks for pointing out.
- “Lines 804 and 805. The abbreviations should be placed immediately below Figure 2”:a. We have added definitions for the abbreviations at the end of the Figure 2 legend. Thank you for the suggestion.